# Relationship of Limb Lengths and Body Composition to Lifting in Weightlifting

**DOI:** 10.3390/ijerph18020756

**Published:** 2021-01-17

**Authors:** Dafnis Vidal Pérez, José Miguel Martínez-Sanz, Alberto Ferriz-Valero, Violeta Gómez-Vicente, Eva Ausó

**Affiliations:** 1Faculty of Health Sciences, University of Alicante, 03690 Alicante, Spain; dafnisvidalperez@gmail.com; 2Research Group on Food and Nutrition (ALINUT), Department of Nursing, Faculty of Health Sciences, University of Alicante, 03690 Alicante, Spain; 3Department of General Didactics and Specific Didactics, Faculty of Education, University of Alicante, 03690 Alicante, Spain; alberto.ferriz@ua.es; 4Department of Optics, Pharmacology and Anatomy, Faculty of Sciences, University of Alicante, 03690 Alicante, Spain; vgvicente@ua.es (V.G.-V.); eva.auso@ua.es (E.A.)

**Keywords:** weightlifter, anthropometry, athletic performance, biomechanics

## Abstract

Weightlifting is a discipline where technique and anthropometric characteristics are essential to achieve the best results in competitions. This study aims to analyse the relationships between body composition, limb length and barbell kinematics in the performance of weightlifters. It consists of an observational and descriptive study of 19 athletes (12 men [28.50 ± 6.37 years old; 84.58 ± 14.11 kg; 176.18 ± 6.85 cm] and 7 women [27.71 ± 6.34 years old; 64.41 ± 7.63 kg; 166.94 ± 4.11 cm]) who met the inclusion criteria. A level I anthropometrist took anthropometric measures according to the methodology of the International Society for the Advancement of Kinanthropometry (ISAK), and the measurement of the barbell velocity was made with the software Kinovea. In terms of body composition, both genders are within the percentage range of fat mass recommended for this sport. In female weightlifters, there is a positive correlation between foot length, maximal velocity in the Snatch (ρ = 0.775, *p* = 0.041), and performance indicator in the Snatch and the Clean & Jerk (ρ = 0.964, *p* < 0.001; ρ = 0.883, *p* = 0.008, respectively). In male weightlifters, a positive correlation between tibial length and average velocity of the barbell in the Snatch is observed (ρ = 0.848, *p* < 0.001). Muscle mass percentage correlates positively with performance indicator in both techniques (ρ = 0.634, *p* = 0.027; ρ = 0.720, *p* = 0.008). Also, the relative length of the upper limb is negatively correlated with the performance indicator (ρ = −0.602, *p* = 0.038). Anthropometry and body composition may facilitate skill acquisition among this sport population, contributing to increase the limited body of scientific knowledge related to weightlifting.

## 1. Introduction

Strength-based sports have been and are still present in every society, being among the oldest and most universal of sports. The maximum strength and the athlete’s anthropometric characteristics are crucial for a successful performance [1]. Weightlifting is a discipline where performance depends on the heaviest load that a competitor can lift in one or in two movements: the snatch (SN) and the clean and jerk (C&J) [1]. In both variants, the goal is to lift a barbell from the ground to overhead, with full extension of the arms. In the SN, the barbell is lifted from the ground and, in one continuous motion, is received in a squat position fixed at arm’s length overhead [2,3,4,5]. According to the change in the knee angle and the height of the barbell, the SN is divided into six phases: the first pull, the transition, the second pull, the turnover under the barbell, the catch phase and the rising from the squat position [2,3,4,5]. By contrast, the C&J is a two-motion lift, existing a pause between these two movements [6]. During the clean, the barbell must be lifted off from the floor to a support position on the anterior aspect of the shoulders in one continuous motion, and in the jerk, the barbell has to be raised overhead and finish with the elbows in a locked out position. There are twelve phases in the C&J, being the first six equal to those described for the snatch and in the last one, or recovery phase, the lifter recovers from the full squat position to prepare for the jerk. The jerk has six more phases, being the first one the start phase, where the lifter and the barbell must become motionless. The following phases are dip, jerk drive, unsupported split under the bar, supported split under the bar, and recovery [7]. Thus, successful performance in this sport is a product of extensive physical and mental preparation that pairs strength and power with technical mastery. It seems obvious to think that this dynamic strength-speed sport requires precise whole-body coordination performed by the weightlifter and a highly impulsive ability and force production [8]. Because the force is directly related to the ability to accelerate an object and to the power that can be produced, sufficient levels of muscular mass, linked to high muscular output, may be considered as the foremost determinant of performance success in elite weightlifters [8]. Back in 1956, Lietzke already reported a strong correlation between body mass and muscle strength in weightlifters [9].

Perhaps, elite powerlifters and weightlifters are the strongest athletes in the world [10], who compete in categories based on age, body weight, and sex. Although many factors would contribute to such strength, it appears likely that the anthropometric profile of these lifters is one of them. For example, light- (≤56 kg) to middle-weight (≤85 kg) male weightlifters’ somatotype is predominantly ectomorph or mesomorph [11], with 5–10% body fat percentages [12], whereas heavy-weight athletes tend to be more endomorphic mesomorphs [11] with body fat percentages over 17% [12]. In female weightlifters, the body fat percentage may double that of male athletes of similar body mass [12]. In comparison with other strength and power athletes, Olympic weightlifters find advantage from being relatively short with short limb segments and large biacromial breadths [13]. Specifically, Fry et al. observed that elite junior Olympic lifters had higher (moderate to large effect size) levels of fat-free mass, a lower percentage of body fat, and a shorter humerus, tibia, and trunk than non-elite junior lifters [13]. Likewise, having shorter lower limbs relative to stature has also been documented in highly skilled male weightlifters [14,15]. However, not all studies point out in the same direction. For instance, Musser et al. found a correlation between having a long trunk and thigh in international professional female weightlifters and their ability to lift a heavier load in the SN [16]. This result agrees with previous studies, providing evidence of the importance of having a long trunk in SN performance in both, females and males [14]. On the contrary, the influence of thigh length differs between genders and, while in females a longer thigh seems to favour performance success [16,17], in males is a disadvantage [15]. Therefore, determining the optimal anthropometric proportions for weightlifting appears difficult because each of the lifts has somewhat specific anthropometric requirements. For example, short arms, which are in general advantageous for weightlifters, can be detrimental to perform the deadlift [18,19].

Another parameter involved in weightlifting success is barbell kinematics, which can also be conditioned by the athlete’s anthropometric characteristics. Some kinematic parameters of weightlifting exercises that have been typically analysed are the barbell trajectory, velocity, and acceleration during the different pull phases of either the SN or the C&J [6,20]. The barbell velocity and acceleration profiles provide important information about the lifter’s technical abilities [21]. However, literature studying and relating barbell kinematics to weightlifters’ anthropometric variables is scarce. In this regard, it is well known that the minimum horizontal barbell displacement is considered an essential element of optimal lifting technique [22], and at least one study has reported a relationship between horizontal barbell displacement and some anthropometric variables (e.g., lower limb length) that was weight class-dependent [16]. In this study, a longer lower limb in female weightlifters correlated with a decreased barbell horizontal movement [16]. Other studies have analysed, together with the barbell velocity, the angular kinematics of the lower limb by measuring and comparing the extension angle and velocity of the lower limb joints (ankle, knee and hip) in junior elite female weightlifters [2,23]. Those studies are limited to the quantification of such variables without ascertaining a possible relationship between them. More recent studies investigating the relationship between barbell kinematics and net joint movements of the limb during the clean have found that the net joint movements produced at the knee and the ankle largely contribute to barbell velocity and acceleration during weightlifting [24].

Despite the latest investigations regarding barbell kinematics and anthropometric parameters, the literature on weightlifting is limited compared with many other Olympic sports, and it seems that both gender and ethnicity may influence anthropometric traits. While investigations in weightlifting are infrequent, studies in the Spanish population are absent. Therefore, given that barbell kinematics are an integral part of the weightlifting technique, the study of its relationship with limbs and segments length, linked to the performance of weightlifting athletes, acquires relevance. Thus, the aim of this work was to describe possible correlations between body composition, anthropometric parameters and barbell kinematics in weightlifting, to assess the importance of anthropometric traits in performance. Consequently, it was initially hypothesised that:

**Hypothesis** **(H1).***Lifters who have the shortest limbs show higher performance on weightlifting, both in males and females*.

**Hypothesis** **(H2).***Lifters with the highest percentage of muscle mass show a higher performance on weightlifting, both in males and females*.

**Hypothesis** **(H3).***Lifters with the shortest limbs present higher speed of the bar (average and peak), both in males and females*.

## 2. Materials and Methods

### 2.1. Design

Observational and descriptive study on the relationship of body composition, extremity lengths and bar kinematics in the performance of weightlifting athletes.

### 2.2. Sample

The subjects were 44 amateur weightlifters (men *n* = 31, and women *n* = 13), of legal age, from the provinces of Alicante, Valencia and Albacete in Spain. All athletes, who competed at the national level, were recruited voluntarily from different weightlifting clubs that agreed to take part in the study and were affiliated to the Spanish Federation of Weightlifting (RFEH, in Spanish). In accordance with the Helsinki Declaration of 2013 and the University of Alicante Research Ethics Committee (permit no. #UA-2019-02-14), the participants who fulfilled the inclusion criteria gave their written consent after being informed about the objectives, the procedures and the methodology of the study.

### 2.3. Inclusion and Exclusion Criteria

All athletes over the age of 18 who reported 2 or more years of experience in the practice of weightlifting, trained a minimum of 3 h per week, were affiliated to the RFEH, participated in at least one competition during the period studied and were able to perform a complete snatch, were eligible to take part in the study. Subjects who did not meet the inclusion criteria, as well as those who suffered from injuries before or during the period covered, were excluded from the study.

According to these criteria, several athletes were discarded from the initial sample: four trained less than 3 h per week; four resulted injured during the period studied; eight abandoned the study, and nine failed to participate in any competition. The final sample consisted of 12 men (28.50 ± 6.37 years old) and 7 women (27.71 ± 6.34 years old).

### 2.4. Materials

The equipment used in the anthropometric measurements included a skinfold caliper (accuracy 0.5 mm, Cescorf, Porto Alegre, Brazil), a soft, thin and inextensible metric tape (accuracy 1 mm, Cescorf), a small and large bone anthropometer (accuracy 1 mm, Cescorf), a segmometer (accuracy 1 mm, Cescorf), a stadiometer (accuracy 1 mm), a weight scale (accuracy 100 g, Omron, Hoofddorp, The Netherlands. ), an anthropometric box (dimensions 40 × 50 × 30 cm^3^), and a cosmetic pencil. All instruments were homologated and properly calibrated. Video records of the lifts performed by weightlifters were captured at 60 Hz using a commercial-grade video camera (Reflex EOS 1200D, Canon, Madrid, Spain). The measurement of the barbell velocity was made with the Kinovea-0.8.15 software (https://www.kinovea.org/) [25].

### 2.5. Procedures

To carry out the study, the following variables were collected with descriptive or statistical purposes: sex (male or female); age; bodyweight category for competition (weight classes); training hours per week; repetition maximum (RM); average velocity and peak velocity during the snatch, clean and jerk; skinfolds; girths; breadths; lengths; body composition; and proportionality indexes. To determine statistical correlations, the value of the load each participant was able to lift was normalized using the load lifted by the first-ranked athlete in the sex- and bodyweight-matched category of the 2019 Weightlifting World Championships [26] as reference (100%) (Snatch Performance Indicator = [load lifted by the subject/load lifted by the first position in the same-sex and bodyweight category of the 2019 Weightlifting World Championships] × 100). These data are reflected and detailed in the results section.

Data were collected from December of 2018 till April of 2019 during the first competition of the season. The anthropometric measurements were carried out at the clubs’ facilities on the day scheduled with the coach and previously to the training sessions, to interfere with the athlete’s routine as little as possible. The protocols and standards of the International Society for the Advancement of Kinanthropometry (ISAK) were followed to get the measures [27]. To ensure this, a level I anthropometrist, accredited by the ISAK, performed the measurements under the supervision of a level III anthropometrist, always taking into consideration the intraobserver technical error of the measurement (TEM) specified by the ISAK (5% for skinfolds and 1% for girths and breadths).

The participants underwent the following measurements: basic measurements (body mass and stature); skinfolds (triceps, subscapular, biceps, iliac crest, supraspinal, abdominal, thigh, calf); girths (relaxed arm, flexed and tensed arm, waist, hip, calf); breadths (humerus, bi-styloid, femur); and lengths (upper limb (Acromiale-Dactylion), arm (Acromiale-Radiale), forearm (Radiale-Stylion), hand (Midstylion-Dactylion), lower limb (Trochanterion height), femur (Trochanterion-Tibiale laterale), calf (Tibiale laterale height), foot (Pternion-Akropodion)), comprising a total of 26 anthropometric variables.

The anthropometric assessment of body composition was based on the evaluation of four components (fat mass (FM) by Yuhasz, muscle mass (MM) by Rose and Guimaraes, bone mass (BM) by Rocha, and residual mass by Würch), as proposed by De Rose and Guimaraes and the Spanish group of Kinanthropometry (GREC) [27,28]; these values were expressed in kg, as well as a percentage of the total body mass. In addition, the following proportionality indexes were calculated: brachial index ([forearm length/arm length] × 100), intermembral index ([upper limb length/lower limb length] × 100), relative length of the upper limb ([upper limb length/stature] × 100), and relative length of the lower limb ([lower limb length/stature] × 100). Table 1 describes the anthropometric characteristics and body composition of the study sample.

To perform the video recording, a digital camera was mounted on a tripod at 1.5 m of height from the ground and 5 m to the right side of the competition platform centre, providing a sagittal plane view of the right side of the athletes and the barbell discs, and matching the side of the body the anthropometric measures were taken from (Figure 1). Once recorded, the videos were uploaded to the App, adjusting the target to the contour of the barbell disc as accurately as possible, in order to minimise errors. To measure the velocity of the barbell during the snatch, we considered as the initial point its lift-off from the ground, and as the final point, the highest peak reached during its vertical displacement prior to the catch position in the squat. To measure the velocity of the barbell during the clean and jerk, these movements were regarded as separate. The initial and the final point of the clean were considered, respectively, the barbell lift-off from the ground and the first contact of the barbell with the athlete’s shoulders (to avoid an incorrect measure due to bouncing). For the jerk, the initial point was the vertical pull when the barbell was driven up off the shoulders, and the final point was the barbell positioned overhead with the athlete’s arms fully extended (Table 2). Each weightlifter completed a one RM.

### 2.6. Statistical Analysis

Data were entered onto an SPSS^®^ Version 24.0.0.0 statistical programme (IBM^®^ International Business Machines Corp., Madrid, Spain) where the mean and standard deviation for each variable were determined in order to describe participants’ characteristics. Spearman’s correlations were used to determine relationships between basic anthropometric measurements (body mass [kg], stature [cm] and limbs length [cm]), body composition measurements (fat mass [%], bone mass [%], muscle mass [%] and fat-free mass index) and performance variables in weightlifting (average velocity [m·s^−1^], maximal velocity [m·s^−1^] and performance indicator [%] in the snatch, clean and jerk. Significance level was set as *p* ≤ 0.05. The strength of the correlation coefficient (ρ) was designated as per Hopkins [29]. A ρ value ranging from 0 to 0.30 or 0 to −0.30 was considered small; 0.31 to 0.49, or −0.31 to −0.49, moderate; 0.50 to 0.69, or −0.50 to −0.69, large; 0.70 to 0.89, or −0.70 to −0.89, very large; and 0.90 to 1, or −0.90 to −1, near perfect for relationship prediction.

## 3. Results

Table 3 presents individual data collection from each subject.

### 3.1. Males

#### 3.1.1. Squat Snatch

When analyzing the data to find possible correlations between anthropometric and kinematic parameters in men performing the SN, we identified the following statistically significant relationships. Firstly, there was a very large, positive relationship between tibial length and average velocity of the barbell (Table 4 and Figure 2). Muscle mass percentage, lower limb length and radial length positively correlated with the performance indicator (large) (Table 4 and Figure 3). Secondly, the relative length of the upper limb negatively correlated with the performance indicator (large).

#### 3.1.2. Clean & Jerk

On the one hand, for the Clean technique, a large positive relationship between the average velocity of the barbell and body mass was detected (Table 4). Also, the average velocity negatively correlated with the bone mass percentage (large). The maximum velocity positively correlated with the fat-free mass index (large).

On the other hand, for the Jerk technique, there was a positive relationship between the average velocity of the barbell and body mass (very large), stature (very large), lower limb length (very large), foot length (large) and upper limb length (large). Additionally, the average velocity negatively correlated with bone mass percentage (large) and the maximum velocity positively correlated with the fat mass percentage (large) (Table 4).

Finally, for the C&J technique, there was a positive relationship between the muscle mass percentage and the performance indicator (very large). Consequently, the fat mass percentage negatively correlated with the performance indicator (large) (Table 4).

### 3.2. Females

#### 3.2.1. Squat Snatch

When analyzing the data to find possible correlations between anthropometric and kinematic parameters in women performing the SN, we identified the following statistically significant relationships. Firstly, positive correlations of the foot length with the maximum velocity of the barbell (near perfect), and with the performance indicator (very large) were detected (Table 5 and Figure 4). Secondly, the hand length positively correlated with the performance indicator as well (very large, Table 5 and Figure 5).

#### 3.2.2. Clean & Jerk

A very large positive relationship between the foot length and the performance indicator was determined for the C&J technique (Table 5).

## 4. Discussion

The purpose of this study was to investigate body composition and the importance of limb length in relation to bar kinematics during the SN and the C&J movements in amateur weightlifters. To our knowledge, there is no research investigating the relationships between weightlifting performance, barbell velocity, and anthropometrics (length) of the upper and lower limbs in amateur weightlifters [2,13,23,24]. The results obtained revealed that both body composition and limb length correlated positively and significantly with bar kinematics. Furthermore, this study also investigated men and women separately to ascertain the possible influence of gender differences on anthropometric parameters such as body mass, stature, muscle mass, bone mass, upper and lower limb lengths, and upper limb to lower limb ratio, in relation to weightlifting performance. Together, the results indicated that taller and heavier male athletes, as well as those who showed a higher proportion of muscle and bone mass, were able to perform better on weightlifting, which translated into a higher average velocity of the barbell.

The average body mass of elite Spanish weightlifters is 75.2 ± 12.80 kg for men and 59.3 ± 9.94 kg for women [30]; being different from the average body mass of men in our sample (84.58 ± 14.11 kg) and similar in the case of women (64.41 ± 7.63 kg). In the case of the stature, the data of the Spanish weightlifters (179.5 ± 8.33 [30] and 172.1 ± 6.3 cm [31] for men; and 166.3 ± 7.39 cm for women) are similar to those of our sample (176.18 ± 6.85 cm for men; and 166.94 ± 4.11 cm for women). Other studies also show similar weights and statures for men and women weightlifters [32,33].

It has been previously reported that a short stature implies an advantage in this sport discipline [30] but, in general terms, the taller weightlifters are also the ones whose body mass is higher [12,32]. Because in weightlifting body mass determines the weight class the athletes compete in, body composition and segments’ length become extremely relevant for successful performance. In both genders, athletes with higher body mass are generally able to lift heavier loads, given that their muscle mass is also greater. However, in terms of relative strength, the shorter a weightlifter is, the higher his strength will be [12,32]. This is because the body is primarily composed of third class levers, therefore, the longer the bony segments, the greater the work and torque the lifter requires to lift the barbell [34,35]. Moreover, weightlifters can be expected to be heavier than age-matched individuals in the general population [32,36]. Compared to jumping or throwing disciplines, the body mass is relevant for explosive strength. In fact, in throwing disciplines higher body mass is a prerequisite for success but, due to the division in weight categories, weightlifters with lighter body mass can excel in competitions [32,37]. Thus, factors such as muscle mass or lean mass in male and female weightlifters influence performance across all weight categories [7].

In all sports, but especially in weightlifting, body composition (fat mass and lean body mass) is a fundamental aspect that needs to be evaluated for athletic development [7]. In the present study, body composition was evaluated according to the four components model: muscle mass, fat mass, bone mass and residual mass [27]. Conforming with this model, the results indicate that both, women and men are within the range of fat mass percentage described for the athletic population [30,38]. The fat mass has to be kept low, but the lean mass and muscle mass must be increased, as they are associated with greater strength and power performances [7]. In fact, it has been shown that the total lean body mass influences the level of strength and power at all ages and weight categories in Olympic weightlifting [39,40]. In this context, fat mass recorded in our sample was within the fat percentage range recommended by Wilmore and Costill (5–12% for men and 10–18% for women) [41]. In other studies that also collect anthropometric data from male weightlifters, similar values of fat mass percentage (10.6 ± 4.5% and 10.1 ± 4.0% versus 9.99 ± 2.45%) can be found [12,31]. In our sample, women showed lower fat mass compared with data from other studies where female individuals showed a similar body mass [12].

Regarding muscular-skeletal development, the literature recommends the use of the fat-free mass index as a body composition metric to assess relative muscularity in athletes. The values of this index in our male sample are comparable to those of the Spanish athletic population (>24 kg lean mass/m^2^), indicating a pronounced muscle development [30]. Similar results have been described for college weightlifting athletes [42]. In the case of females, our values are higher than those of the averaged Spanish athletes (19.52 kg lean mass/m^2^ versus <18.5 kg lean mass/m^2^), which indicates that women in our sample have an average muscle development compared with a low muscle development of the Spanish athletes [30]. This index can be used to guide nutritional and exercise interventions, predict athletic performance, and provide information regarding the potential for further fat-free mass accretion in male athletes [42].

Regarding the limbs in men, we observed that, as one would have expected, the longer they were (and therefore, the longer their segments were), the higher the average velocity of the barbell. Specifically, we detected a highly significant positive correlation between the total length of the limbs and their respective body segments (tibia and foot in the lower limb; radio in the upper limb) with the SN and Jerk average velocity (Table 4). In contrast, in women we only found a correlation of the hand and the foot lengths with the maximum velocity of the barbell and the performance indicator, in both weightlifting modalities (SN and C&J) (Table 5). Therefore, our results suggest that there are specific differences between genders on the potential influence of the upper and lower limb lengths on the barbell velocity. Nevertheless, these results may have been influenced by the small sample size of women (*n* = 7) in this study.

In agreement with previous studies, the comparison of limb lengths and their corresponding segments between our sample and data of elite weightlifters, reveals smaller lengths in amateur athletes [13,16]. In this sense, it is well known, that a shorter stature and limb lengths of weightlifters provide mechanical advantages when lifting heavy loads by reducing the mechanical torque and the vertical distance that the barbell must be displaced [7,13]. However, a larger skeletal structure (frame) would be advantageous for the accumulation of muscle mass [18], so the longer upper and lower limbs of our weightlifters would favour an increase in muscle mass. Concerning this, the average percentage of muscle mass in the sample was superior to 50% (51.03 ± 3.47 for men and 48.82 ± 2.98 for women) showing a high and significant positive correlation with performance in the SN (0.634) and C&K (0.720). Additionally, taller athletes will need to generate more force and do more work to complete the SN and C&J movements, due to the bar requirement of travelling further and maintaining a higher average velocity through the duration of the lift.

Finally, the limited experience and technical skills of our amateur weightlifters could have led to an incorrect technique, influencing the barbell velocity. According to Kipp and Harris, the velocity and acceleration profiles provide important information about the lifter’s technical abilities [24], as well as about the velocity required to successfully complete the SN or Clean [43]. In addition, at the early learning stages for novice athletes, another important factor to take into account is the posture of the weightlifter at the beginning of the lift, because this position allows inertia of the bar to be overcome using the larger muscles that cross the hips [44,45]. It is widely accepted that starting positions will differ between weightlifters and it is their anthropometric characteristics, which predominantly govern how they should best set up. The knee and ankle joint flexion angle have a greater relative importance in bar kinematics during weightlifting [23,24,46] and taller weightlifters with longer lower limbs require a greater knee flexion angle to get into an optimal start position, helping an optimal technical development and bar kinematics (Bartonietz et al., 1996). Previous studies revealed that increased ankle dorsiflexion was contrary to the good-starting position [47], as well as the stability of the weightlifter and the barbell trajectory that can interfere in the outcome of a SN attempt [22,43,44]. The restricted experience and skills, as well as the lack of confidence and stability of the amateur weightlifters possibly conditionate the adoption of a body posture during the catch where the knee flexion is above 90° and the ankle dorsiflexion increases, gaining stability but negatively affecting the technique [23,24,46,47]. However, our results are in disagreement to Lockie et al., where no correlations were found between men’s leg and arm lengths and barbell velocity in strength-based sports such as the deadlift, while significant positive relationships between women’s upper and lower limb with average barbell velocity were found [48]. It has been proposed that those who are taller, or have a longer femur, may have longer resistance moment arms [49].

In women, our results showed a highly positive significant correlation between hand length and performance in the SN. The hand is the final link along the kinetic chain, where the generated forces and torques are transferred to the barbell, hence the functional importance of the hand to weightlifting performance [50]. It is well known that strength-based sports require certain degree of handgrip strength (see review Crorin et al. [51]). Our results are in the same direction to Fry et al., who found a relationship between high handgrip strength and weightlifting ability [13].

### Limitations

Some limitations must be considered when interpreting the results of this study. One of them is the sample size of the weightlifters recruited, because there were only 12 males and seven females, who showed considerable within-group variation in stature and body mass, and this could influence the statistical analysis carried out. Anyway, small sample sizes characterise the studies carried out so far on this subject in strength and weightlifting sports. On the other hand, the scientific literature found on this topic was scarce, restricting the comparison and discussion of our results. Moreover, one of the variables more analysed in the few existing studies was the strength and the angles of the different segments adopted in the initial posture with each of the techniques, which has not been analysed in the present study, making it difficult to compare our results. So, this study contributes to the limited body of scientific knowledge related to weightlifting. It is also a step towards understanding the previously unexplored relationship between the athlete’s anthropometry, mainly limb lengths, and barbell movement.

## 5. Conclusions

The results from this study provide information about anthropometric factors that influence lifting mechanics in weightlifting (women and men). So, anthropometry and body composition could help and facilitate training and skill acquisition for the athletes of this discipline, contributing to increasing the limited body of scientific knowledge related to weightlifting. According with our results, weightlifters with greater body mass, stature, muscle mass, bone mass, fat-free mass, as well as longer limb lengths correlated with higher average values of barbell velocity in squat SN and C&J. However, given the small sample size and the results obtained, further studies are necessary to confirm the present findings. Weightlifting coaches, athletes, health and sports professionals can benefit from a better understanding of the association between anthropometry and barbell velocity to optimise technique based on individual body dimensions.

## Figures and Tables

**Figure 1 ijerph-18-00756-f001:**
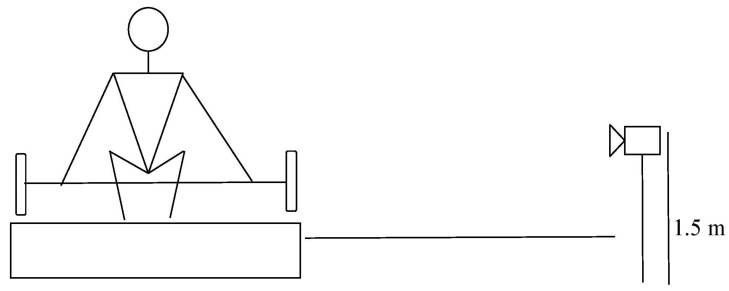
Arrangement of the camera for video recordings.

**Figure 2 ijerph-18-00756-f002:**
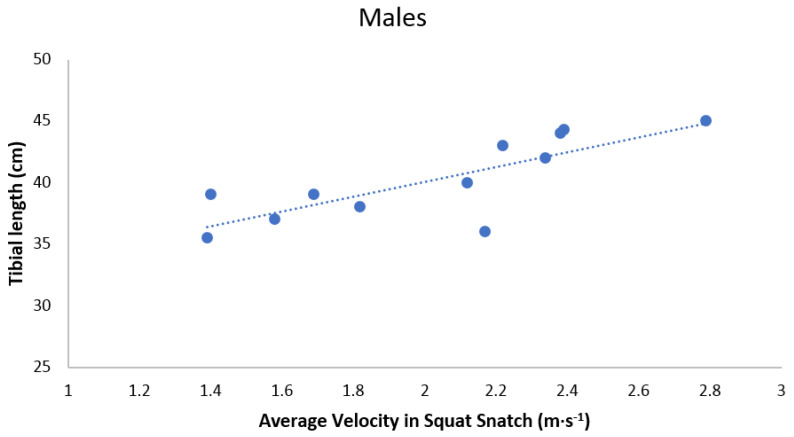
Scatterplot relationship between tibial length (cm) and average velocity in Squat Snatch in males.

**Figure 3 ijerph-18-00756-f003:**
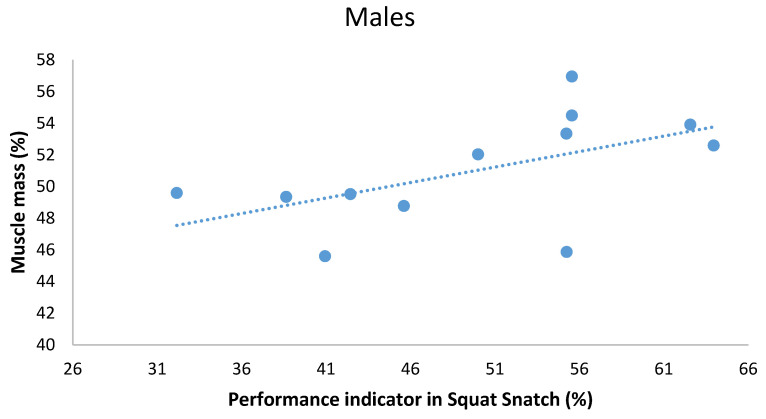
Scatterplot relationship between muscle mass (%) and performance indicator (%) in Squat Snatch in males.

**Figure 4 ijerph-18-00756-f004:**
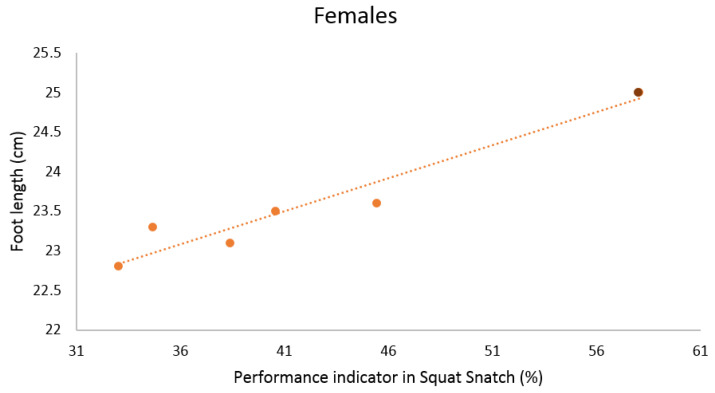
Scatterplot relationship between foot length (cm) and performance indicator (%) in Squat Snatch in females.

**Figure 5 ijerph-18-00756-f005:**
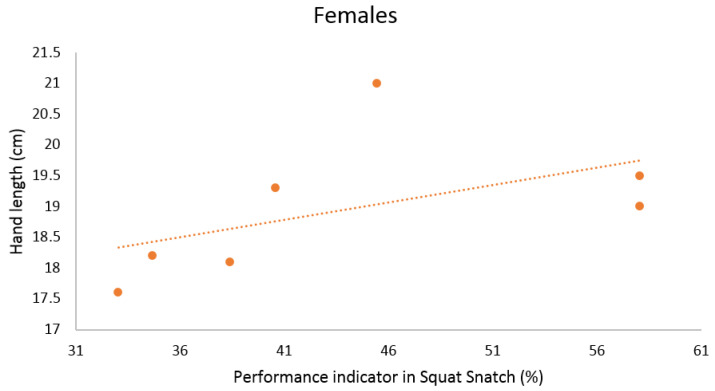
Scatterplot relationship between hand length (cm) and performance indicator in Squat Snatch in females.

**Table 1 ijerph-18-00756-t001:** Anthropometric characteristics, body composition and proportionality indexes of the study sample disaggregated by sex. Values represent mean ± standard deviation.

		Sample					
		Men			Women		
Sport Modality	Weightlifting	*n* = 12			*n* = 7		
**Basic measures**	Body mass (kg)	84.6	±	14.1	64.4	±	7.6
	Stature (cm)	176.2	±	6.9	166.9	±	4.1
**Skinfolds (mm)**	Triceps	8.2	±	3.5	13.2	±	5.5
	Subscapular	12.4	±	6.3	9.7	±	2.6
	Biceps	4.5	±	2.0.	5.5	±	2.4
	Iliac crest	16.5	±	4.7	14.3	±	3.8
	Supraspinale	10.2	±	5.6	9.0	±	3.55
	Abdominal	14.6	±	5.38	14.1	±	7.0
	Thigh	12.8	±	4.64	15.2	±	3.14
	Calf	7.3	±	3.25	13.3	±	6.5
**Girths (cm)**	Arm (relaxed)	34.5	±	2.9	29.7	±	2.6
	Arm (flexed and tensed)	37.1	±	2.4	30.6	±	2.2
	Waist (minimum)	87.4	±	7.6	72.5	±	4.3
	Hip (maximum)	100.8	±	6.4	98.2	±	4.6
	Calf (maximum)	38.8	±	3.1	35.6	±	2.9
**Breadths (mm)**	Humerus	7.1	±	0.4	6.7	±	1.0
	Bi-styloid	5.8	±	0.4	5.2	±	0.2
	Femur	9.9	±	0.7	8.9	±	0.4
**Body composition**	% Body fat	10.0.	±	2.5	15.2	±	3.0
	% Muscle mass	51.6	±	3.0	48.8	±	3.0
	% Bone mass	14.4	±	1.4	15.1	±	1.1
	Fat mass (kg)	8.7	±	3.5	9.9	±	2.6
	Muscle mass (kg)	43.5	±	6.5	31.4	±	3.9
	Bone mass (kg)	12.1	±	1.8	9.7	±	0.9
**Lengths (cm)**	Lower limb	94.5	±	3.9	92.7	±	2.87
	Femur	48.1	±	2.4	48.8	±	1.4
	Calf	40.2	±	3.4	36.6	±	2.6
	Foot	26.8	±	1.4	23.8	±	0.9
	Upper limb	78.7	±	3.2	73.5	±	1.9
	Arm	34.0.	±	1.3	32.0	±	1.3
	Foream	24.3	±	1.1	22.2	±	1.1
	Hand	20.3	±	1.3	19.0	±	1.1
**Anthropometric indexes**	Fat-free mass Index	24.4	±	2.1	19.5	±	1.4
	Brachial index	71.5	±	3.4	69.4	±	3.3
	Intermembral index	83.3	±	2.0	79.3	±	3.3

**Table 2 ijerph-18-00756-t002:** Description of the movements corresponding to the snatch (left side) and the clean and jerk (right side).

Movement	One Motion (Snatch)		Two Motions (Clean and Jerk)
Initial point of snatch: barbell lift-off from the ground	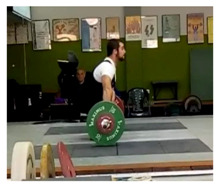	Initial point of clean: barbell lift-off from the ground	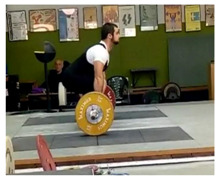
Final point of snatch: barbell peaks in vertical trajectory (maximum height)	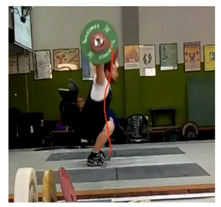	Final point of clean: first contact of the barbell with the athlete’s shoulders in two movements	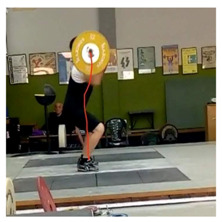
		Initial point of jerk: barbell lift-off from the athlete’s shoulders	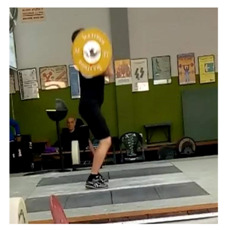
		Final point of jerk: full extension of the athlete’s arms and legs split in the final position	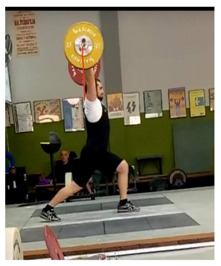

**Table 3 ijerph-18-00756-t003:** Individual characteristics and data collection from each subject.

Sa	BM	S	THW	CC	Load (kg)		Average Velocity (m·s^−1^)			Peak Velocity (m·s^−1^)		
					Snatch	C&J	Snatch	Clean	Jerk	Snatch	Clean	Jerk
**Males**												
**M1**	70.0	170.0	3	S 73	83 ^A^	108 ^B^	2.17	1.53	1.10	3.40	2.64	2.28
**M2**	76.0	166.0	3	S 81	55 ^C^	70 ^D^	1.69	1.41	1.18	3.56	3.00	3.23
**M3**	75.5	176.4	3	S 81	70 ^C^	90 ^D^	1.39	1.58	1.54	3.50	3.08	3.01
**M4**	75.3	173.0	4	S 81	78 ^C^	105 ^D^	2.38	1.14	1.29	3.59	2.38	2.49
**M5**	77.7	171.0	5	S 81	95 ^C^	115 ^D^	2.12	1.80	1.53	3.50	2.71	2.93
**M6**	79.7	168.0	5	S 81	95 ^C^	130 ^D^	1.40	1.77	1.17	3.27	3.02	2.07
**M7**	81.5	179.8	5	S 89	107 ^E^	139 ^F^	1.58	1.68	1.61	3.22	2.63	2.71
**M8**	88.7	184.1	3	S 89	73 ^E^	95 ^F^	2.39	2.18	1.76	3.40	2.82	2.60
**M9**	88.7	181.1	5	S 89	95 ^E^	140 ^F^	2.22	2.11	1.57	3.56	3.55	2.81
**M10**	82.5	177.8	5	S 89	110 ^E^	130 ^F^	2.34	2.10	1.58	3.65	3.26	2.66
**M11**	96.4	178.0	6	S 102	100 ^G^	105 ^H^	1.82	2.04	1.37	3.46	2.85	3.06
**M12**	123.0	189.0	4	S + 109	85 ^I^	108 ^J^	2.79	1.83	1.79	4.17	3.10	2.93
**Females**												
**F1**	52.5	160.0	6	S 55	45 ^K^	63 ^L^	1.49	1.40	1.64	3.49	3.38	2.79
**F2**	57.4	163.4	5	S 59	43 ^M^	57 ^N^	3.66	3.20	3.38	2.12	1.21	1.64
**F3**	64.0	165.2	3	S 71	37 ^P^	45 ^Q^	3.36	2.70	2.79	2.30	1.60	1.70
**F4**	66.0	170.0	3	S 71	43 ^P^	55 ^Q^	3.71	3.12	2.79	1.57	1.14	1.64
**F5**	66.5	170.3	5	S 71	65 ^P^	90 ^Q^	1.66	1.34	1.78	3.60	3.33	2.98
**F6**	68.5	170.1	5	S 71	65 ^P^	85 ^Q^	1.82	1.27	1.92	3.70	3.28	3.16
**F7**	76.0	170.2	3	S 76	43 ^R^	50 ^S^	3.83	2.71	2.55	1.68	1.38	1.38

Sa = Sample; BM = Body Mass (kg); S = Stature (cm); THW = Training hours per week; CC = Competition Category (weight classes); M1 = Male 1; M2 = Male 2; etc.; F1 = Female 1; F2 = Female 2; etc. ^A^. Shi Zhiyong (166 kg); ^B^. Shi Zhiyong (197 kg); ^C^. Lu Xiaojung (171 kg); ^D^. Lu Xiaojung (207 kg); ^E^. Revaz Davitadze (172 kg); ^F^. Toshiki Yamamoto (208 kg); ^G^. Jin Yun-seong (181 kg); ^H^. Reza Dehdar (219 kg); ^I^. Lasha Talakhadze (220 kg); ^J^. Lasha Talakhadze (264 kg); ^K^. Zhang Wanqiong (99 kg) ^L^. Liao Quiyun (129 kg) ^M^. Kuo Hsing-chun (106 kg) ^N^. Kuo Hsing-chun (140 kg) ^P^. Katherine Nye (112 kg) ^Q^. Katherine Nye (136 kg); ^R^. Rim Jong-sim (124 kg) ^S^. Zhang Wangli (153 kg).

**Table 4 ijerph-18-00756-t004:** Spearman’s correlation coefficient and bilateral significance (in brackets) between basic anthropometric measurements, body composition measurements and performance variables in the snatch, clean and jerk in males (*n* = 12).

	Snatch			Clean		Jerk		C&J
	AV	MV	PI	AV	MV	AV	MV	PI
**Basic measures**								
Body mass (kg)	0.182	0.218	0.091	0.606	0.501	0.739	0.435	−0.056
	(0.571)	(0.495)	(0.778)	(0.037)	(0.097)	(0.006)	(0.157)	(0.863)
Stature (cm)	0.573	0.264	0.186	0.552	0.098	0.832	0.175	0.049
	(0.051)	(0.408)	(0.564)	(0.063)	(0.762)	(0.001)	(0.586)	(0.880)
**Girths (cm)**								
Arm (relaxed)	0.364	0.214	0.095	0.385	0.140	0.161	−0.074	−0.070
	(0.245)	(0.503)	(0.770)	(0.217)	(0.665)	(0.618)	(0.820)	(0.829)
Arm (flexed and tensed)	0.238	0.141	0.151	0.483	0.238	0.336	0.011	−0.077
	(0.457)	(0.663)	(0.640)	(0.112)	(0.457)	(0.286)	(0.974)	(0.812)
Waist (minimum)	0.035	0.221	−0.168	0.238	0.413	0.280	0.182	−0.399
	(0.914)	(0.489)	(0.601)	(0.457)	(0.183)	(0.379)	(0.571)	(0.199)
Hip (maximum)	0.091	0.267	−0.459	−0.168	0.049	0.070	0.203	−0.497
	(0.779)	(0.401)	(0.134)	(0.602)	(0.880)	(0.829)	(0.527)	(0.101)
Calf (maximum)	0.254	0.319	−0.051	0.109	0.215	0.264	−0.145	−0.190
	(0.427)	(0.313)	(0.875)	(0.736)	(0.503)	(0.407)	(0.654)	(0.554)
**Body composition**								
Fat mass (%)	0.168	0.457	−0.557	−0.014	0.161	0.266	0.588	−0.692
	(0.602)	(0.135)	(0.060)	(0.966)	(0.618)	(0.404)	(0.044)	(0.013)
Muscle mass (%)	−0.154	−0.316	0.634	0.154	−0.021	−0.084	−0.350	0.720
	(0.633)	(0.316)	(0.027)	(0.633)	(0.948)	(0.795)	(0.264)	(0.008)
Bone mass (%)	−0.252	0.025	−0.392	−0.629	−0.196	−0.692	−0.025	−0.385
	(0.430)	(0.939)	(0.207)	(0.028)	(0.542)	(0.013)	(0.940)	(0.217)
Fat-free mass Index	−0.133	0.239	0.046	0.336	0.608	0.420	0.438	−0.203
	(0.681)	(0.454)	(0.888)	(0.286)	(0.036)	(0.175)	(0.155)	(0.527)
**Lengths (cm)**								
Lower limb (LL)	0.637	0.359	0.163	0.459	0.137	0.767	0.023	0.172
	(0.026)	(0.252)	(0.612)	(0.134)	(0.672)	(0.004)	(0.944)	(0.594)
Relative lower limb	0.245	0.243	−0.186	−0.538	−0.245	−0.322	−0.193	0
	(0.443)	(0.448)	(0.564)	(0.071)	(0.443)	(0.308)	(0.549)	(1)
Femur	0.007	0.141	−0.072	−0.189	−0.119	0.323	0.348	−0.330
	(0.983)	(0.662)	(0.824)	(0.555)	(0.712)	(0.306)	(0.268)	(0.295)
Calf	0.848	0.562	−0.181	0.448	0.207	0.494	−0.147	−0.088
	(<0.001)	(0.057)	(0.574)	(0.144)	(0.519)	(0.103)	(0.648)	(0.787)
Foot	0.574	0.178	−0.075	0.252	−0.011	0.616	−0.037	−0.077
	(0.051)	(0.580)	(0.816)	(0.429)	(0.974)	(0.033)	(0.909)	(0.812)
Upper limb (UL)	0.466	0.149	−0.037	0.269	−0.244	0.696	0.227	−0.194
	(0.126)	(0.644)	(0.909)	(0.399)	(0.445)	(0.012)	(0.479)	(0.545)
Relative upper limb	−0.070	0.148	−0.602	−0.517	−0.084	−0.287	0.039	−0.566
	(0.829)	(0.647)	(0.038)	(0.085)	(0.795)	(0.366)	(0.905)	(0.055)
Arm	0.128	0.360	−0.270	−0.277	−0.213	0.489	0.426	−0.376
	(0.693)	(0.250)	(0.396)	(0.384)	(0.507)	(0.106)	(0.167)	(0.228)
Forearm	0.667	0.239	0.030	0.452	0.035	0.367	−0.246	−0.046
	(0.018)	(0.454)	(0.926)	(0.141)	(0.913)	(0.241)	(0.442)	(0.887)
Hand	0.399	0.059	−0.375	0.180	−0.071	0.452	0.287	−0.491
	(0.198)	(0.856)	(0.229)	(0.575)	(0.827)	(0.14)	(0.366)	(0.105)
Brachial index	0.245	−0.213	0.333	0.550	0.151	−0.182	−0.477	0.340
	(0.442)	(0.506)	(0.290)	(0.064)	(0.640)	(0.571)	(0.117)	(0.280)
Intermembral index	−0.245	−0.060	−0.480	−0.126	0.063	0.070	0.273	−0.559
	(0.443)	(0.854)	(0.114)	(0.697)	(0.846)	(0.829)	(0.390)	(0.059)

AV = average velocity; MV = maximum velocity; PI = Performance Indicator; C&J = Clean and Jerk.

**Table 5 ijerph-18-00756-t005:** Spearman’s correlation coefficient and bilateral significance (in brackets) between basic anthropometric measurements, body composition measurements and performance variables in the snatch, clean and jerk in females (*n* = 7).

	Snatch			Clean		Jerk		C&J
	AV	MV	PI	AV	MV	AV	MV	PI
**Basic measures**								
Body mass (kg)	0.393	0	0.054	−0.250	−0.107	−0.126	−0.072	−0.071
	(0.383)	(1)	(0.908)	(0.589)	(0.819)	(0.788)	(0.878)	(0.879)
Stature (cm)	0.394	−0.039	0.119	−0.236	−0.256	−0.03	0.030	0.020
	(0.382)	(0.933)	(0.799)	(0.610)	(0.579)	(0.949)	(0.949)	(0.967)
**Girths (cm)**								
Arm (relaxed)	0.523	−0.324	−0.418	0.198	−0.270	0.227	−0.455	−0.414
	(0.229)	(0.478)	(0.350)	(0.670	(0.558)	(0.624)	(0.306)	(0.355)
Arm (flexed and tensed)	−0.071	0	−0.180	−0.071	0.321	−0.270	−0.162	−0.107
	(0.879)	(1)	(0.699)	(0.879	(0.482)	(0.558)	(0.728)	(0.819)
Waist (minimum)	0.536	−0.321	−0.270	0.036	−0.286	0.054	−0.342	−0.321
	(0.215)	(0.482)	(0.558)	(0.939)	(0.535)	(0.908)	(0.452)	(0.482)
Hip (maximum)	0.071	0.214	−0.090	−0.357	0.107	−0.054	0.162	0
	(0.879)	(0.645)	(0.848)	(0.432)	(0.819)	(0.908)	(0.728)	(1)
Calf (maximum)	0.357	0.143	−0.018	−0.357	−0.036	−0.090	0.018	−0.143
	(0.432)	(0.760)	(0.969)	(0.432)	(0.939)	(0.848)	(0.969)	(0.760)
**Body composition**								
Fat mass (%)	0.286	−0.286	−0.198	0.143	−0.321	0.234	−0.162	−0.071
	(0.535)	(0.535)	(0.670)	(0.760)	(0.482)	(0.613)	(0.728)	(0.879)
Muscle mass (%)	−0.214	0.250	0.090	−0.214	0.464	−0.432	0.018	−0.071
	(0.645)	(0.589)	(0.848)	(0.645)	(0.294)	(0.333)	(0.969)	(0.879)
Bone mass (%)	−0.214	−0.071	0.198	0.071	−0.107	0	0.198	0.214
	(0.645)	(0.879)	(0.670)	(0.879)	(0.819)	(1)	(0.670)	(0.645)
Fat-free mass Index	0.429	0.071	−0.216	−0.286	−0.107	0.036	−0.054	−0.321
	(0.337)	(0.879)	(0.641)	(0.535)	(0.819)	(0.939)	(0.908)	(0.482)
**Lengths (cm)**								
Lower limb (LL)	0.162	−0.216	0.255	0.414	−0.180	0.118	−0.336	0.180
	(0.728)	(0.641)	(0.582)	(0.355)	(0.699)	(0.801)	(0.461)	(0.699)
Relative lower limb	−0.143	−0.179	0.144	0.429	0.071	−0.018	−0.288	0.143
	(0.760)	(0.702)	(0.758)	(0.337)	(0.879)	(0.969)	(0.531)	(0.760)
Femur	0.505	−0.580	−0.179	0.617	−0.356	0.179	−0.736	−0.337
	(0.247)	(0.172)	(0.700)	(0.140)	(0.434)	(0.700)	(0.059)	(0.460)
Calf	0.198	−0.090	0.400	0.306	−0.414	0.373	−0.055	0.378
	(0.670)	(0.848)	(0.374)	(0.504)	(0.355)	(0.410)	(0.908)	(0.403)
Foot	−0.631	0.775	0.964	−0.667	0.631	−0.655	0.709	0.883
	(0.129)	(0.041)	(<0.001)	(0.102)	(0.129)	(0.111)	(0.074)	(0.008)
Upper limb (UL)	0.234	0.252	0.427	−0.090	−0.234	0.236	0.145	0.288
	(0.613)	(0.585)	(0.339)	(0.848)	(0.613)	(0.610)	(0.756)	(0.531)
Relative upper limb	−0.414	0.613	0.391	−0.306	0.342	−0.055	0.473	0.360
	(0.355)	(0.144)	(0.386)	(0.504)	(0.452)	(0.908)	(0.284)	(0.427)
Arm	0.371	0	0.318	−0.074	−0.445	0.224	0.112	0.185
	(0.413)	(1)	(0.487)	(0.875)	(0.317)	(0.629)	(0.811)	(0.691)
Forearm	0.252	0.144	−0.318	0.036	−0.234	0.564	0.009	−0.252
	(0.585)	(0.758)	(0.487)	(0.939)	(0.613)	(0.188)	(0.985)	(0.585)
Hand	−0.679	0.500	0.757	−0.321	0.643	−0.631	0.414	0.714
	(0.094)	(0.253)	(0.049)	(0.482)	(0.119)	(0.129)	(0.355)	(0.071)
Brachial index	0.214	0	−0.450	0.214	−0.107	0.468	−0.234	−0.357
	(0.645)	(1)	(0.310)	(0.645)	(0.819)	(0.289)	(0.613)	(0.432)
Intermembral index	0.214	0.036	−0.288	−0.214	−0.179	0.162	0.126	−0.321
	(0.645	(0.939)	(0.531)	(0.645)	(0.702)	(0.728)	(0.788)	(0.482)

AV = average velocity; MV = maximum velocity; PI = Performance Indicator; C&J = Clean and Jerk.

## Data Availability

The data presented in this study are available in the tables of this article. The data presented in this study are available on request from the corresponding author.

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
