# Peer review of "Relationship of Limb Lengths and Body Composition to Lifting in Weightlifting"

_ijerph, 2021, doi:10.3390/ijerph18020756_

Round 1
Reviewer 1 Report
Thank you for the opportunity to review the manuscript entitled: "Relationship of Limb Lengths and Body Composition to Lifting in Weightlifting". The research problem undertaken by the authors is very interesting and may be of great importance for the coaching practice. However, at the moment the article needs significant revisions in order for its acceptance in IJERPH to be considered.
Major concerns:
The number of participants is extremely low, (especially in the group of women) for a study analyzing the correlation between variables. It is worth emphasizing that this is not a group of elite athletes that are difficult to recruit. It is not enough to list this as a study limitation. In addition, the Authors mention that lifters are considerably different between weight classes, maybe more accurate should be to analyze correlation within weight class? Moreover, in the group of women can be seen two outliers, which can mess up your analysis. In my opinion, this is a serious problem that prevents this study from being accepted.
Minor:
Line 16: there is nothing about nutrition in this study, "nutrition" seems unnecessary
Line 18-19: more details about the participants should be provided, as well as in Materials and Methods section, e.g. mean and SD of body mass, BMI or/and BF, height, years, training experience, strength level etc.).
Line 22-23: this sentence seems unnecessary if you provide more details about participants
Line 24: the velocity of the technique sounds odd to me. you were measuring barbell velocity
Line 23-28: there are a lack o significance and correlation values
Line 31: keywords should not duplicate words that are in the title
Line 36: this statement need reference
Line 69: Provide years of this reference
Line 73: lack of "et al."
Materials and Methods
as I mentioned before, more details about participants are needed (e.g. weight classes?).
Line 144: please provide more details about this software, is it validated? What's about reliability? Moreover, there should be more information about velocity measurement, on what %1RM velocity was measured and why?
Line 150-155: please provide a rationale for normalization instead of the absolute value of the maximum strength
Line 215: have you studied the normality and homogeneity of the data?
Overall the discussion seems written correctly, however in addition to revising critically discussion in order to the issues raised above, the authors must soften their conclusions.
Line 293: "Weightlifting is a discipline where performance depends on the heaviest load that a competitor can lift" therefore please explain why lifters should be interested in the average velocity of the bar? not on external load?
Line 294-299: what is the purpose of this paragraph?
Author Response
We thank the reviewers for their careful evaluation of our manuscript. We have made modifications to our manuscript according to the corrections and concerns expressed in their comments.
We attach a cover letter to each of the comments made by the reviewer

Reviewer 2 Report
I would like to thank you for submitting and give me the opportunity to review this work about the relationships among anthropometry, body composition and performance in weightlifters. I hope my comments will help to improve the quality of the manuscript in some way.
The paper is mostly well written, with a correct English grammar and in general gives enough information in every section of the manuscript. Nevertheless, some questions and concerns need to be answered and corrected before the formal acceptance of the manuscript.
In first place, it is necessary to add some data in the abstract that give specific information about the relationships between the different variables studied.
In line 20 change increasing by increase.
Load data used by the participants is not displayed. In the method they mention an indicator of normalization of performance, but data of these parameters are not shown throughout the results of the article. From my point of view, these data are relevant, since the load used in weightlifting will be relevant in the speed of the movement and therefore will affect the correlations made. For this reason, it is necessary to explain in more depth how the load selection was carried out for each participant and show that data.
Authors also need to add barbell velocity data in both techniques.
My main concern is that the technique used by weightlifters to lift the weight is very relevant to achieve optimal performance and the greatest possible speed in the movement of the bar. However, in this study no information is provided on some parameters that could be relevant, such as the angles of the different segments adopted in the initial posture with each of the techniques (for example the knee angle). Taking into account that a recording was made from the sagittal plane to measure the speed of the barbell, a kinematic analysis of the position in 2D could have been carried out to assess whether the effect of these variables is greater than the effect of the anthropometry of the participants. In addition, the authors discuss their results of anthropometric variables with previous studies in which the kinematics of the position were analyzed, when they have not measured these variables, so although there may be a certain relationship, they are not comparable parameters.
Author Response

(The authors gave the same response as above.)

Reviewer 3 Report
Weightlifting is one of the most commonly used training aids for both trained and untrained athletes. Therefore, studies that contribute to the understanding of the relationship between biological assumptions and performance are current and necessary. Therefore, the assessed work can be evaluated positively. The text will need to be amended before it can be published. From the content data, it will be necessary to add in the hypothesis which anthropometric parameters will affect the speed of the dumbbell's movement and, of course, it is necessary to evaluate the established hypothesis. When presenting the monitored persons, it is necessary to state in which period the survey was monitored - meaning from the point of view of body mass adjustment. What prediction equations were used to calculate BC parameters? Have they been adapted to Spanish conditions? What was the measurement error of the input parameters? It does not make sense to present the obtained data to two decimal places. I do not understand the parameter average weight of weightlifters. This means that this is the average of all weight categories. If so, all weight categories were evenly represented here. If not, this parameter does not make sense. How was the execution technique evaluated? As the authors explain the positive significant correlation between hand length and performance in the SN in women, which is in contrast to the literature data, where athletes with shorter limbs are preferred. The discussion will need to be more focused on explaining the results obtained. Similarly, the conclusions of the study need to be more concrete. Formally, it is necessary to summarize specific numerical results and conclusions. I recommend shortening the general detailed description of the implementation of an individual experiment in the introduction of the study.
Author Response

(The authors gave the same response as above.)

Round 2
Reviewer 1 Report
The authors really improved the quality of the manuscript. I am satisfied and think that the paper deserves to be published in IJERPH.
Reviewer 2 Report
Authors have answered properly to all the suggestions made in the previous review round, significantly improving the quality of their work. In this sense, the article is ready for publication.
Reviewer 3 Report
The text was modified according to the author's comments, although the final recommendations would need to be further specified. On the other hand, the data, despite the unrepresentative set, provide the reader with interesting information and are good for further follow-up research. In my opinion, a study in this form is suitable for publication.